# GIFT: Learning Transformation-Invariant Dense Visual Descriptors via Group CNNs

**Yuan Liu  Zehong Shen  Zhixuan Lin  Sida Peng  Hujun Bao*  Xiaowei Zhou***

State Key Lab of CAD&CG, ZJU-Sensetime Joint Lab of 3D Vision, Zhejiang University

## Abstract

Finding local correspondences between images with different viewpoints requires local descriptors that are robust against geometric transformations. An approach for transformation invariance is to integrate out the transformations by pooling the features extracted from transformed versions of an image. However, the feature pooling may sacrifice the distinctiveness of the resulting descriptors. In this paper, we introduce a novel visual descriptor named Group Invariant Feature Transform (GIFT), which is both discriminative and robust to geometric transformations. The key idea is that the features extracted from the transformed versions of an image can be viewed as a function defined on the group of the transformations. Instead of feature pooling, we use group convolutions to exploit underlying structures of the extracted features on the group, resulting in descriptors that are both discriminative and provably invariant to the group of transformations. Extensive experiments show that GIFT outperforms state-of-the-art methods on several benchmark datasets and practically improves the performance of relative pose estimation.

## 1  Introduction

Establishing local feature correspondences between images is a fundamental problem in many computer vision tasks such as structure from motion [21], visual localization [17], SLAM [43], image stitching [5] and image retrieval [47]. Finding reliable correspondences requires image descriptors that effectively encode distinctive image patterns while being invariant to geometric and photometric image transformations caused by viewpoint and illumination changes.

To achieve the invariance to viewpoints, traditional methods [36, 37] use patch detectors [33, 39] to extract transformation covariant local patches which are then normalized for transformation invariance. Then, invariant descriptors can be extracted on the detected local patches. However, a typical image may have very few pixels for which viewpoint covariant patches can be reliably detected [22]. Also, "hand-crafted" detectors such as DoG [36] and Affine-Harris [39] are sensitive to image artifacts and lighting conditions. Reliably detecting covariant regions is still an open problem [29, 11] and a performance bottleneck in the traditional pipeline of correspondence estimation.

Instead of relying on a sparse set of covariant patches, some recent works [20, 7, 11] propose to extract dense descriptors by feeding the whole image into a convolutional neural network (CNN) and constructing pixel-wise descriptors from the feature maps of the CNN. However, the CNN-based descriptors are usually sensitive to viewpoint changes as convolutions are inherently not invariant to geometric transformations. While augmenting training data with warped images improves the robustness of learned features, the invariance is not guaranteed and a larger network is typically required to fit the augmented datasets.

In order to explicitly improve invariance to geometric transformations, some works [60, 55, 22] resort to integrating out the transformations by pooling the features extracted from transformed versions of

the original images. But the distinctiveness of extracted features may degenerate due to the pooling operation.

In this paper, we propose a novel CNN-based dense descriptor, named Group Invariant Feature Transform (GIFT), which is both discriminative and invariant to a group of transformations. The key idea is that, if an image is regarded as a function defined on the translation group, the CNN features extracted from multiple transformed images can be treated as a function defined on the transformation group. Analogous to local image patterns, such features on the group also have discriminative patterns, which are neglected by the previous methods that use pooling for invariance. We argue that exploiting underlying structures of the group features is essential for building discriminative descriptors. It can be theoretically demonstrated that transforming the input image with any element in the group results in a permutation of the group features. Such a permutation preserves local structures of the group features. Thus, we propose to use group convolutions to encode the local structures of the group features, resulting in feature representations that are not only discriminative but equivariant to the transformations in the group. Finally, the intermediate representations are bilinearly pooled to obtain provably invariant descriptors. This transformation-invariant dense descriptor simplifies correspondence estimation as detecting covariant patches can be avoided. Without needs for patch detectors, the proposed descriptor can be incorporated with any interest point detector for sparse feature matching or even a uniformly sampled grid for dense matching.

We evaluate the performance of GIFT on the HPSequence [1, 30] dataset and the SUN3D [59] dataset for correspondence estimation. The results show that GIFT outperforms both of the traditional descriptors and recent learned descriptors. We further demonstrate the robustness of GIFT to extremely large scale and orientation changes on several new datasets. The current unoptimized implementation of GIFT runs at ∼15 fps on a GTX 1080 Ti GPU, which is sufficiently fast for practical applications.

## 2 Related work

Existing pipelines for feature matching usually rely on a feature detector and a feature descriptor. Feature detectors [33, 36, 39] detect local patches which are covariant to geometric transformations brought by viewpoint changes. Then, invariant descriptors can be extracted on the normalized local patches via traditional patch descriptors [36, 6, 48, 3] or deep metric learning based patch descriptors [63, 19, 40, 52, 37, 2, 18, 23, 64]. The robustness of detectors can be guaranteed theoretically, e.g., by the scale-space theory [34]. However, a typical image often have very few pixels for which viewpoint covariant patches may be reliably detected [22]. The scarcity of reliably detected patches becomes a performance bottleneck in the traditional pipeline of correspondence estimation. Some recent works [45, 61, 29, 41, 64, 12, 62] try to learn such viewpoint covariant patch detectors by CNNs. However, the definition of a canonical scale or orientation is ambiguous. Detecting a consistent scale or orientation for every pixel remains challenging.

To alleviate the dependency on detectors, A-SIFT [42] warps original image patches by affine transformations and exhaustively searches for the best match. Some other methods [60, 22, 55, 13] follow similar pipelines but pool features extracted from these transformed patches to obtain invariant descriptors. GIFT also transforms images, but instead of using feature pooling, it applies group convolutions to further exploit the underlying structures of features extracted from the group of transformed images to retain distinctiveness of the resulting descriptors.

**Feature map based descriptor.** Descriptors can also be directly extracted from feature maps of CNNs [20, 11, 7]. However, CNNs are not invariant to geometric transformations naturally. The common strategy to make CNNs invariant to geometric transformations is to augment the training data with such transformations. However, data augmentation cannot guarantee the invariance on unseen data. The Universal Correspondence Network (UCN) [7] uses a convolutional spatial transformer [26] in the network to normalize the local patches to a canonical shape. However, learning an invariant spatial transformer is as difficult as learning a viewpoint covariant detector. Our method also uses CNNs to extract features on transformed images but applies subsequent group convolutions to construct transformation-invariant descriptors.

**Equivariant or invariant CNNs.** Some recent works [8, 38, 28, 10, 58, 15, 27, 9, 28, 57, 38, 24, 14, 16, 4] design special architectures to make CNNs equivariant to specific transformations. The most related work is the Group Equivariant CNN [8] which uses group convolution and subgroup pooling

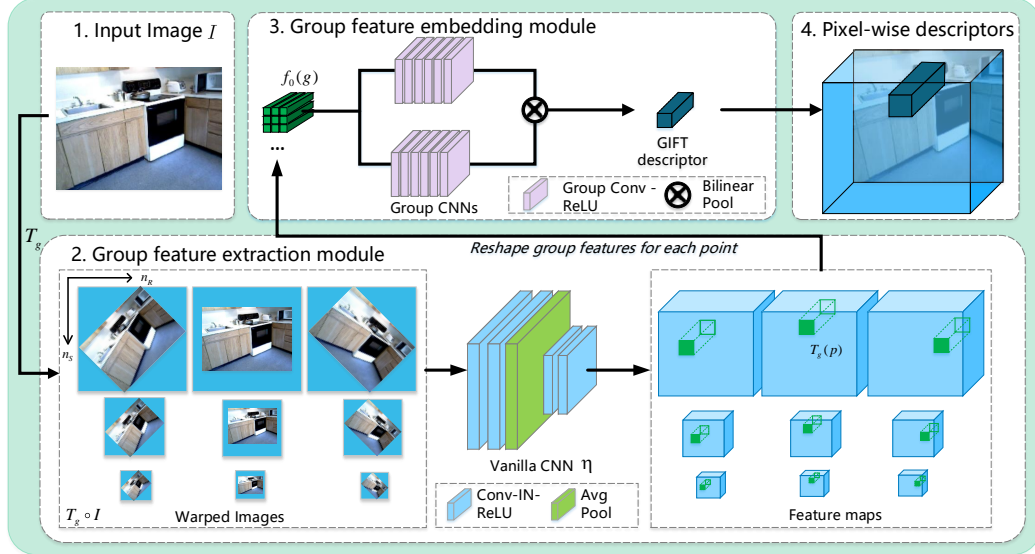

Figure 1: Pipeline. The input image is warped with different transformations and fed into a vanilla CNN to extract group features. Then the group features for each interest point are further processed by two group CNNs and a bilinear pooling operator to obtain final GIFT descriptors.

to learn equivariant feature representations. It applies group convolutions directly on a large group which is the product of the translation group and the geometric transformation group. In contrast, GIFT uses a vanilla CNN to process images, which can be regarded as features defined on the translation group, and separate group CNNs to process the features on the geometric transformation group, which results in a more efficient model than the original Group Equivariant CNN.

## 3 Method

**Preliminary**. Assuming the observed 3D surfaces are smooth, the transformation between corresponding image patches under different viewpoints is approximately in the affine group. In this paper, we only consider its subgroup $G$ which consists of rotations and scaling. The key intermediate feature representation in the pipeline of GIFT is a map $f : G \to \mathbb{R}^n$ from the group $G$ to a feature space $\mathbb{R}^n$, which is referred to as *group feature*.

**Overview**. As illustrated in Fig. 1, the proposed method consists of two modules: group feature extraction and group feature embedding. Group feature extraction module takes an image $I$ as input, warps the image with a grid of sampled elements in $G$, separately feeds the warped images through a vanilla CNN, and outputs a set of feature maps where each feature map corresponds to an element in $G$. For any interest point $p$ in the image, a feature vector can be extracted from each feature map. The feature vectors corresponding to $p$ in all the feature maps form a group feature $f_0 : G \to \mathbb{R}^{n_0}$. Next, the group feature embedding module embeds the group feature $f_0$ of every interest point to two features $f_{l,\alpha}$ and $f_{l,\beta}$ by two group CNNs, both of which have $l$ group convolution layers. Finally, $f_{l,\alpha}$ and $f_{l,\beta}$ are pooled by a bilinear pooling operator [32] to obtain a GIFT descriptor $d$.

### 3.1 Group feature extraction

Given an input image $I$ and a point $p = (x, y)$ on the image, this module aims to extract a transformation-equivariant group feature $f_0 : G \to \mathbb{R}^{n_0}$ on this point $p$. To get the feature vector $f_0(g)$ on a specific transformation $g \in G$, we begin with transforming the input image $I$ with $g$. Then, we process the transformed image $g \circ I$ with a vanilla CNN $\eta$. The output feature map is denoted by $\eta(T_g \circ I)$. Since the image is transformed, the corresponding location of $p$ on the output feature map also changes into $T_g(p)$. We use the feature vector locating at $T_g(p)$ on the feature map $\eta(T_g \circ I)$ as the value of $f_0(g)$. Because the coordinates of $T_g(p)$ may not be integers, we apply a bilinear interpolation $\phi$ to get the feature vector on it. The whole process can be expressed by,

$$f_0(g) = \phi(\eta(T_g \circ I), T_g(p)). \tag{1}$$

The extracted group feature $f_0$ is equivariant to transformations in the group, as illustrated in Fig. 2.

**Lemma 1.** *The group feature of a point $p$ in an image $I$ extracted by Eq. (1) is denoted by $f$. If the image is transformed by an element $h \in G$ and the group feature extracted at the corresponding point $T_h(p)$ in this transformed image is denoted by $f'$, then for any $g \in G$, $f'(g) = \phi(\eta(T_g \circ T_h \circ I), T_g(T_h(p))) = \phi(\eta(T_{gh} \circ I), T_{gh}(p)) = f(gh)$, which means that the transformation of the input image results in a permutation of the group feature.* $\square$

Lemma 1 provides a novel and strict criterion for matching two feature points. Traditional methods usually detect a canonical scale and orientation for an interest point in each view and match points across views by descriptors extracted at the canonical scale and orientation. This can be interpreted as, *if two points are matched, then there exists a $g \in G$ and $g' \in G$ such that $f(g) = f'(g')$*. However, the canonical $g$ and $g'$ are ambiguous and hard to detect reliably. Lemma 1 shows that, *if two points are matched, then there exists an $h \in G$ such that for all $g \in G$, $f'(g) = f(gh)$*. In other words, the group features of two matched points are related by a permutation. This provides a strict matching criterion between two group features. Even though $h$ can hardly be determined when extracting descriptors, the permutation caused by $h$ preserves structures of group features and only changes their locations. Encoding local structures of group features allows us to construct distinctive and transformation-invariant descriptors.

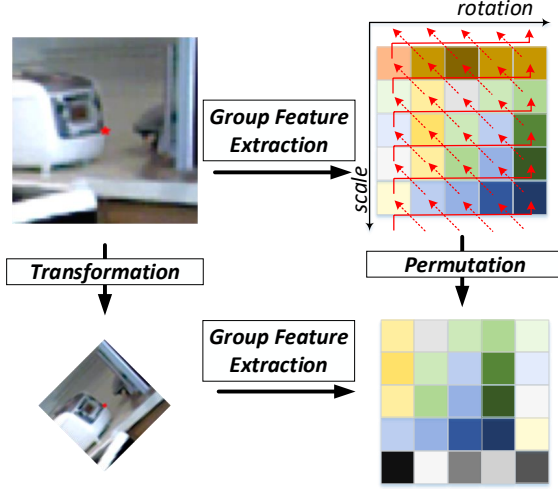

Figure 2: The scaling and rotation of an image (left) result in the permutation of the feature maps defined on the scaling and rotation group (right). The red arrows illustrate the directions of the permutation.

## 3.2 Group convolution layer

After group feature extraction, we apply the discrete group convolution originally proposed in [8] to encode local structures of group features, which is defined as

$$[f_l(g)]_i = \sigma \left( \sum_{h \in H} f_{l-1}^T(hg) W_i(h) + b_i \right), \tag{2}$$

where $f_l$ and $f_{l-1}$ are group features of the layer $l$ and the layer $l-1$ respectively, $[\cdot]_i$ means the $i$-th dimension of the vector, $g$ and $h$ are elements in the group, $H \subset G$ is a set of transformations around the identity transformation, $W$ are learnable parameters which are defined on $H$, $b_i$ is a bias term and $\sigma$ is a non-linear activation function. If $G$ is the 2D translation group, the group convolution in Eq. (2) becomes the conventional 2D convolution. Similar to the conventional CNNs that are able to encode local patterns of images, the group CNNs are able to encode local structures of group features. For more discussions about the relationship between the group convolution and the conventional convolution, please refer to [8].

The group convolution actually preserves the equivariance of group features:

**Lemma 2.** *In Eq. (2), if $f_{l-1}$ is equivariant to transformations in $G$ as stated in Lemma 1, then $f_l$ is also equivariant to transformations in $G$.*

The proof of Lemma 2 is in the supplementary material. With Lemma 2, we can stack multiple group convolution layers to construct group CNNs which are able to encode local structures of group features while maintaining the equivariance property.

## 3.3 Group bilinear pooling

In GIFT, we actually construct two group CNNs $\alpha$ and $\beta$, both of which consist of $l$ group convolution layers, to process the input group feature $f_0$. The outputs of two group CNNs are denoted by $f_{l,\alpha}$

and $f_{l,\beta}$, respectively. Finally, we obtain the GIFT descriptor $d$ by applying the bilinear pooling operator [32] to $f_{l,\alpha}$ and $f_{l,\beta}$, which can be described as

$$d_{i,j} = \int_G [f_{l,\alpha}(g)]_i [f_{l,\beta}(g)]_j dg, \tag{3}$$

where $d_{i,j}$ is an element of feature vector $d$. Based on Lemma 1 and Lemma 2, we can prove the invariance of GIFT as stated in Proposition 1. The proof is given in the supplementary material.

**Proposition 1.** *Let $d$ denote the GIFT descriptor of an interest point in an image. If the image is transformed by any transformation $h \in G$ and the GIFT descriptor extracted at the corresponding point in the transformed image is denoted by $d'$, then $d' = d$.*

In fact, many pooling operators such as average pooling and max pooling can achieve such invariance. We adopt bilinear pooling for two reasons. First, it collects second-order statistics of features and thus produces more informative descriptors. Second, it can be shown that the statistics used in many previous methods for invariant descriptors [22, 55, 60] can be written as special forms of bilinear pooling, as proved in supplementary material. So the proposed GIFT is a more generalized form compared to these methods.

### 3.4 Implementation details

**Sampling from the group**. Due to limited computational resources, we sample a range of elements in $G$ to compute group features. We sample evenly in the scale group $S$ and the rotation group $R$ separately. The unit transformations are defined as 1/4 downsampling and 45 degree clockwise rotation and denoted by $s$ and $r$, respectively. Then, the sampled elements in the group $G$ form a grid $\{(s^i, r^j) | i, j \in \mathbb{Z}\}$. Considering computational complexity, we choose $n_s = 5$ scales ranging from $s^0$ to $s^4$ and $n_r = 5$ orientations ranging from $r^{-2}$ to $r^2$. In this case, the group feature of an interest point is a tensor $f \in \mathbb{R}^{n_s \times n_r \times n}$ where $n$ is the dimension of the feature space.

Due to the discrete sampling, Lemma 1 and Lemma 2 don't rigorously hold near the boundary of the selected range. But empirical results show that this boundary effect will not obviously affect the final matching performance if the scale and rotation changes are in a reasonable range.

**Bilinear pooling**. The integral in the Eq. (3) is approximated by the summation over the sampled group elements. Suppose the output group features of two group CNNs are denoted by $f_{l,\alpha} \in \mathbb{R}^{n_s \times n_r \times n_\alpha}$ and $f_{l,\beta} \in \mathbb{R}^{n_s \times n_r \times n_\beta}$, respectively, and reshaped as two matrices $\tilde{f}_{l,\alpha} \in \mathbb{R}^{n_g \times n_\alpha}$ and $\tilde{f}_{l,\beta} \in \mathbb{R}^{n_g \times n_\beta}$, where $n_g = n_s \times n_r$. Then, the GIFT descriptor $d \in \mathbb{R}^{n_\alpha \times n_\beta}$ can be written as

$$d = \tilde{f}_{l,\alpha}^T \tilde{f}_{l,\beta}. \tag{4}$$

**Network architecture**. The vanilla CNN has four convolution layers and an average pooling layer to enlarge receptive fields. In the vanilla CNN, we use instance normalization [53] instead of batch normalization [25]. The output feature dimension $n_0$ of the vanilla CNN is 32. In both group CNNs, $H$ defined in Eq. (2) is $\{r, r^{-1}, s, s^{-1}, rs, rs^{-1}, r^{-1}s, r^{-1}s^{-1}, e\}$, where $e$ is the identity transformation. ReLU [44] is used as the nonlinear activation function. The number of group convolution layers $l = 1$ in ablation studies and $l = 6$ in subsequent comparisons to state-of-the-art methods. The output feature dimensions $n_\alpha$ and $n_\beta$ of two group CNNs are 8 and 16 respectively, which results in a 128-dimensional descriptor after bilinear pooling. The output descriptors are L2-normalized so that $\|d\|_2 = 1$.

**Loss function**. The model is trained by minimizing a triplet loss [49] defined by

$$\ell = max\left(\|d_a - d_p\|_2 - \|d_a - d_n\|_2 + \gamma, 0\right), \tag{5}$$

where $d_a$, $d_p$ and $d_n$ are descriptors of an anchor point in an image, its true match in the other image, and a false match selected by hard negative mining, respectively. The margin $\gamma$ is set to 0.5 in all experiments. The hard negative mining is a modified version of that proposed in [7].

## 4 Experiments

### 4.1 Datasets and Metrics

**HPSequences** [1, 30] is a dataset that contains 580 image pairs for evaluation which can be divided into two splits, namely Illum-HP and View-HP. Illum-HP contains only illumination changes

while View-HP contains mainly viewpoint changes. The viewpoint changes in the View-HP cause homography transformations because all observed objects are planar.

**SUN3D** [59] is a dataset that contains 500 image pairs of indoor scenes. The observed objects are not planar so that it introduces self-occlusion and perspective distortion, which are commonly-considered challenges in correspondence estimation.

**ES-\*** and **ER-\***. To fully evaluate the correspondence estimation performance under extreme scale and orientation changes, we create extreme scale (ES) and extreme rotation (ER) datasets by artificially scaling and rotating the images in HPSequences and SUN3D. For a pair of images, we manually add large orientation or scale changes to the second image. The range of rotation angle is $[-\pi, \pi]$. The range of scaling factor is $[2.83, 4] \bigcup [0.25, 0.354]$. Examples are shown in Fig. 3.

**MVS dataset** [51] contains six image sequences of outdoor scenes. All images have accurate ground-truth camera poses which are used to evaluate the descriptors for relative pose estimation.

**Training data**. The proposed GIFT is trained on a synthetic dataset. We randomly sample images from MS-COCO [31] and warp images with reasonable homographies defined in Superpoint [11] to construct image pairs for training. When evaluating on the task of relative pose estimation, we further finetune GIFT on the GL3D [50] dataset which contains real image pairs with ground truth correspondences given by a standard Structure-from-Motion (SfM) pipeline.

**Metrics**. To quantify the performance of correspondence estimation, we use Percentage of Correctly Matched Keypoints (PCK) [35, 65], which is defined as the ratio between the number of correct matches and the total number of interest points. All matches are found by nearest-neighbor search. A matched point is declared being correct if it is within five pixels from the ground truth location. To evaluate relative pose estimation, we use the rotation error as the metric, which is defined as the angle of $R_{err} = R_{pr} \cdot R_{gt}^T$ in the axis-angle form, where $R_{pr}$ is the estimated rotation and $R_{gt}$ is the ground truth rotation. All testing images are resized to $480 \times 360$ in all experiments.

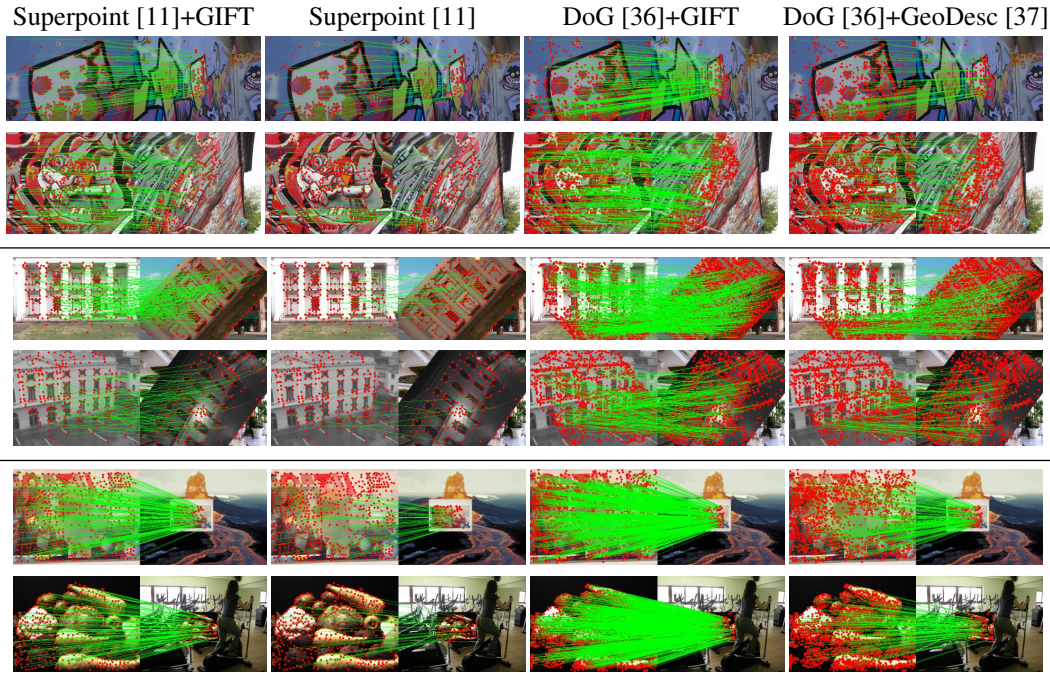

Figure 3: Visualization of estimated correspondences on HPSequences (first two rows), ER-HP (middle two rows) and ES-HP (last two rows). The first two columns use keypoints detected by Superpoint [11] and the last two columns use keypoints detected by DoG [36].

## 4.2 Ablation study

We conduct ablation studies on HPSequence, ES-HP and ER-HP in three aspects, namely comparison to baseline models, choice of pooling operators and different numbers of group convolution layers. In all ablation studies, we use the keypoints detected by Superpoint [11] as interest points for evaluation.

|         | VCNN  | GFC   | GAS   | GIFT-1 |
|---------|-------|-------|-------|--------|
| Illum-HP | 59.15 | **60.63** | 59.2  | 59.61  |
| View-HP  | 61.7  | 62.5  | 62.2  | **63.71** |
| ES-HP    | 14.9  | 16.58 | 18.28 | **21.74** |
| ER-HP    | 28.86 | 26.89 | 30.72 | **39.68** |

Table 1: PCK of different baseline models and GIFT-1.

|         | avg   | max   | subspace | bilinear |
|---------|-------|-------|----------|----------|
| Illum-HP | 57.72 | 54.31 | 47.21    | **59.61** |
| View-HP  | 62.52 | 58.16 | 49.36    | **63.71** |
| ES-HP    | 19.08 | 19.37 | 14.85    | **21.74** |
| ER-HP    | 36.15 | 32.57 | 29.12    | **39.68** |

Table 2: PCK of models using different pooling operators.

We denote the proposed method by GIFT-$l$ where $l$ means the number of group convolution layers. Architectures of compared models can be found in the supplementary material. All tested models are trained with the same loss function and training data.

**Baseline models**. We consider three baseline models which all produce 128-dimensional descriptors, namely Vanilla CNN (VCNN), Group Fully Connected network (GFC) and Group Attention Selection network (GAS). VCNN has four vanilla convolution layers with three average pooling layers and outputs a 128-channel feature map. Descriptors are directly interpolated from the output feature map. GFC and GAS have the same group feature extraction module as GIFT. GFC replaces the group CNN in GIFT-1 with a two-layer fully connected network. GAS is similar to the model proposed in [54], which tries to learn attention weights by CNNs to select a scale for each keypoint. GAS first transforms input group features to 128-dimension by a $1 \times 1$ convolution layer. Then, it applies a two-layer fully connected network on the input group feature to produce $n_r \times n_s$ attention weights. Finally, GAS uses the average of 128-dimensional embedded group features weighted by the attention weights as descriptors.

Table 1 summarizes results of the proposed method and other baseline models. The proposed method achieves the best performance on all datasets except Illum-HP. The Illum-HP dataset contains no viewpoint changes, which means that there is no permutation between the group features of two matched points. Then, the GFC model which directly compares the elements of two group features achieves a better performance. Compared to baseline models, the significant improvements of GIFT-1 on ES-HP and ER-HP demonstrate the benefit of the proposed method to deal with large scale and orientation changes.

**Pooling operators**. To illustrate the necessity of bilinear pooling, we test other three commonly-used pooling operators, namely average pooling, max pooling and subspace pooling [22, 55, 56]. For all these models, we apply the same group feature extraction module as GIFT. For average pooling and max pooling, the input group feature is fed into group CNNs to produce 128-dimensional group features which are subsequently pooled with average pooling or max pooling to construct descriptors. For subspace pooling, we use a group CNN to produce a feature map with 16 channels, which results in 256-dimensional descriptors after subspace pooling. Results are listed in Table 2 which shows that the bilinear pooling outperforms all other pooling operators.

**Number of group convolution layers**. To further demonstrate the effect of group convolution layers, we test on different numbers of group convolution layers. All models use the same vanilla CNN but different group CNNs with 1, 3 or 6 group convolution layers. The results in the Table 3 show that the performance increases with the number of group convolution layers. In subsequent experiments, we use GIFT-6 as the default model and denote it with GIFT for short.

|         | GIFT-1 | GIFT-3 | GIFT-6 |
|---------|--------|--------|--------|
| Illum-HP | 59.61  | 61.33  | **62.49** |
| View-HP  | 63.71  | 64.91  | **67.15** |
| ES-HP    | 21.74  | 23.9   | **27.29** |
| ER-HP    | 39.68  | 43.37  | **48.93** |

Table 3: PCK of GIFT using different numbers of group convolution layers.

## 4.3 Comparison with state-of-the-art methods

We compare the proposed GIFT with three state-of-the-art methods, namely Superpoint [11], GeoDesc [37] and LF-Net [45]. For all methods, we use their released pretrained models for comparison. Superpoint [11] localizes keypoints and interpolates descriptors of these keypoints directly on a feature map of a vanilla CNN. GeoDesc [37] is a state-of-the-art patch descriptor which is usually incorporated with DoG detector for correspondence estimation. LF-Net [45] provides a complete pipeline of feature detection and description. The detector network of LF-Net not only

| detector | Superpoint [11] | | DoG [36] | | | LF-Net [45] | |
|---|---|---|---|---|---|---|---|
| descriptor | GIFT | Superpoint | GIFT | SIFT | GeoDesc | GIFT | LF-Net |
| dataset | | [11] | | [36] | [37] | | [45] |
| Illum-HP | **62.49** | 61.13 | **56.58** | 28.38 | 34.41 | **52.17** | 34.55 |
| View-HP | **67.15** | 53.66 | **62.53** | 34.33 | 42.75 | **15.93** | 1.22 |
| SUN3D | **27.32** | 26.4 | **19.97** | 15.2 | 14.53 | **21.73** | 12.93 |
| ES-HP | **27.29** | 12.16 | **22.07** | 18.25 | 19.63 | **7.89** | 0.3 |
| ER-HP | **48.93** | 24.77 | **44.44** | 29.39 | 37.36 | **12.50** | 0.05 |
| ES-SUN3D | **12.37** | 5.94 | **7.40** | 4.09 | 3.42 | **7.61** | 0.55 |
| ER-SUN3D | **22.29** | 14.01 | **15.77** | 15.16 | 15.39 | **15.98** | 10.59 |

Table 4: PCK of GIFT and the state-of-the-art methods on HPSequences, SUN3D and the extreme rotation (*ER-*) and scaling (*ES-*) datasets.

Reference       GIFT       VCNN       Daisy [47]

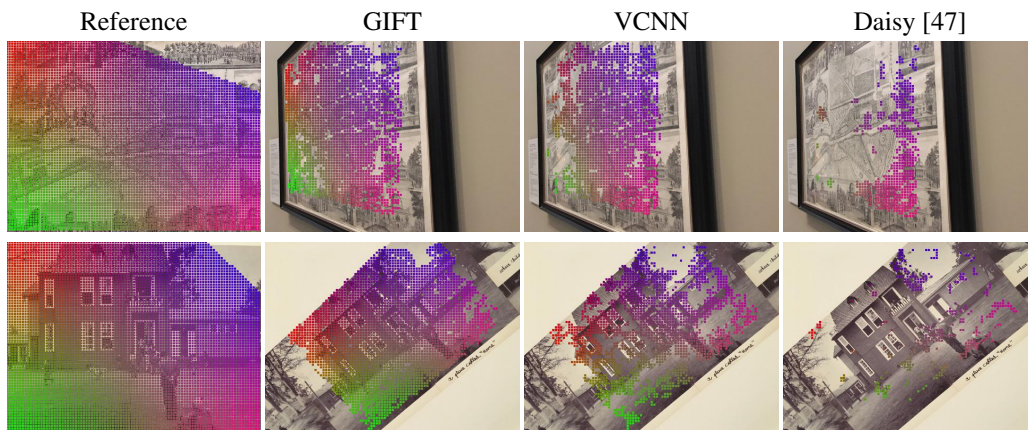

Figure 4: Visualization of estimated dense correspondences. Matched points are drawn with the same color in the reference and query images. Only correctly estimated correspondences are drawn.

localizes keypoints but also estimates their scales and orientations. Then the local patches are fed into the descriptor network to generate descriptors. For fair comparison, we use the same keypoints as the compared method for evaluation. Results are summarized in Table 4, which shows that GIFT outperforms all other state-of-the-art methods. Qualitative results are shown in Fig. 3.

To further validate the robustness of GIFT to scaling and rotation, we add synthetic scaling and rotation to images in HPatches and report the matching performances under different scaling and rotations. The results are plotted in Fig. 5, which show that the PCK of GIFT drops slowly with the increase of scaling and rotation.

## 4.4 Performance for dense correspondence estimation

We also evaluate GIFT for the task of dense correspondence estimation on HPSequence, ES-HP and ER-HP. The quantitative results are listed in Table 5 and qualitative results are shown in Fig. 4. The proposed GIFT outperforms the baseline Vanilla CNN and the traditional method Daisy [47], which demonstrates the ability of GIFT for dense correspondence estimation.

| | GIFT | VCNN | Daisy [47] |
|---|---|---|---|
| Illum-HP | **27.82** | 26.96 | 17.08 |
| View-HP | **37.92** | 32.92 | 19.6 |
| ES-HP | **12.52** | 4.64 | 1.05 |
| ER-HP | **26.61** | 14.02 | 5.69 |

Table 5: PCK of dense correspondence estimation.

## 4.5 Performance for relative pose estimation.

We also evaluate GIFT for the task of relative pose estimation of image pairs on the MVS dataset [51]. For a pair of images, we estimate the relative pose of cameras by matching descriptors and computing

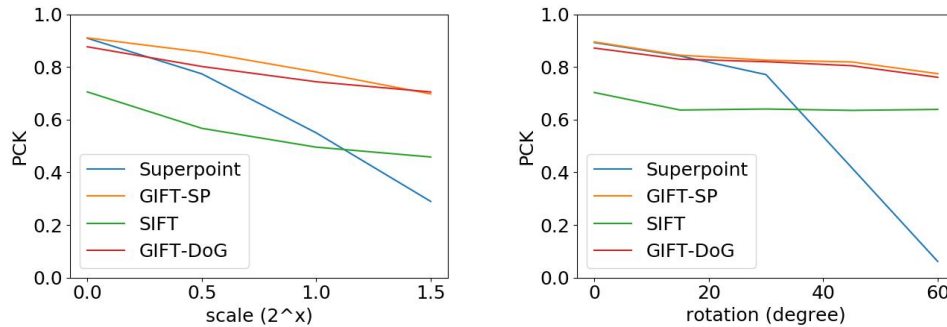

Figure 5: PCKs on the HPatches dataset as scaling and rotation increase. GIFT-SP uses Superpoint as the detector while GIFT-DoG uses DoG as the detector.

essential matrix. Since the estimated translations are up-to-scale, we only evaluate the estimated rotations using the metric of rotation error as mentioned in Section 4.1. We further finetune GIFT on the outdoor GL3D dataset [50] and denote the finetuned model with GIFT-F. The results are listed in Table 6. GIFT-F outperforms all other methods on most sequences, which demonstrates the applicability of GIFT to real computer vision tasks.

| Detector | DoG [36] | | | Superpoint [11] | | |
|---|---|---|---|---|---|---|
| Descriptor | GIFT | GIFT-F | SIFT | GIFT | GIFT-F | Superpoint |
| Sequence | | | [36] | | | [11] |
| Herz-Jesus-P8 | 0.656 | **0.582** | 0.662 | **0.848** | 0.942 | 1.072 |
| Herz-Jesus-P25 | 4.968 | **2.756** | 5.296 | 4.484 | **2.891** | 2.87 |
| Fountain-P11 | 0.821 | 1.268 | **0.587** | 1.331 | **1.046** | 1.071 |
| Entry-P10 | 1.368 | **1.259** | 3.844 | 1.915 | **1.059** | 1.076 |
| Castle-P30 | 3.431 | **1.741** | 2.706 | 1.526 | **1.501** | 1.588 |
| Castle-P19 | **1.887** | 1.991 | 3.018 | 1.739 | **1.500** | 1.814 |
| Average | 2.189 | **1.600** | 2.686 | 1.974 | **1.490** | 1.583 |

Table 6: Rotation error (°) of relative pose estimation on the MVS Dataset [51].

## 4.6 Running time

Given a $480 \times 360$ image and randomly-distributed 1024 interest points in the image, the PyTorch [46] implementation of GIFT-6 costs about 65.2 ms on a desktop with an Intel i7 3.7GHz CPU and a GTX 1080 Ti GPU. Specifically, it takes 32.5 ms for image warping, 27.5 ms for processing all warped images with the vanilla CNN and 5.2 ms for group feature embedding by the group CNNs.

## 5 Conclusion

We introduced a novel dense descriptor named GIFT with provable invariance to a certain group of transformations. We showed that the group features, which are extracted on the transformed images, contain structures which are stable under the transformations and discriminative among different interest points. We adopt group CNNs to encode such structures and applied bilinear pooling to construct transformation-invariant descriptors. We reported state-of-the-art performance on the task of correspondence estimation on the HPSequence dataset, the SUN3D dataset and several new datasets with extreme scale and orientation changes.

**Acknowledgement**. The authors would like to acknowledge support from NSFC (No. 61806176), Fundamental Research Funds for the Central Universities and ZJU-SenseTime Joint Lab of 3D Vision.

## Footnotes

*Corresponding authors: {xzhou,bao}@cad.zju.edu.cn. Project page: https://zju3dv.github.io/GIFT.

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
