[Supplementary Material · 2019_NIPS_Invariant_Descriptor_Supp_final.pdf]

# Supplementary Material:
# GIFT: Learning Transformation-Invariant Dense Visual Descriptors via Group CNNs

**Yuan Liu   Zehong Shen   Zhixuan Lin   Sida Peng   Hujun Bao**$^*$   **Xiaowei Zhou**$^*$

State Key Lab of CAD&CG, ZJU-Sensetime Joint Lab of 3D Vision, Zhejiang University

## 1   Proof of Lemma 2 and Proposition 1

**Proof of Lemma 2**. The input group feature $f_{l-1}$ has the equivariance defined in Lemma 1, which means that transforming the input image $I$ with $h' \in G$ results in a group feature $f'$ satisfying $f'_{l-1}(g) = f_{l-1}(gh')$. Processing $f'_{l-1}$ by a group convolution, the output group feature $f'_l(g)$ is $[f'_l(g)]_i = \sigma\left(\sum_{h \in H} f'_{l-1}(hg)W_i(h) + b_i\right) = \sigma\left(\sum_{h \in H} f_{l-1}(h(gh'))W_i(h) + b_i\right) = [f_l(gh')]_i$.
□

**Proof of Proposition 1**. Based on Lemma 1 and Lemma 2, both the outputs of two group CNNs $f_{l,\alpha}$ and $f_{l,\beta}$ are equivariant to the transformation of image, which means transforming the input image $I$ with $h' \in G$ results in group features $f'_{l,\alpha}$ and $f'_{l,\beta}$ which satisfy $f'_{l,\alpha}(g) = f_{l,\alpha}(gh')$ and $f'_{l,\beta}(g) = f_{l,\beta}(gh')$ respectively. The bilinear pooling of the $f'_{l,\alpha}$ and $f'_{l,\beta}$ is defined as $d'_{i,j} = \int_G [f'_{l,\alpha}(g)]_i [f'_{l,\beta}(g)]_j dg = \int_G [f_{l,\alpha}(gh')]_i [f_{l,\beta}(gh')]_j dg$. Then replacing $gh'$ with $g'$ results in $d'_{i,j} = \int_G [f_{l,\alpha}(g')]_i [f_{l,\beta}(g')]_j dg' = d_{i,j}$.
□

## 2   Bilinear forms of methods [5, 3, 1]

**Subspace pooling [3, 1]**. The local descriptors proposed in the [3, 1] are extracted by singular value decomposition, which is denoted as subspace pooling in [4]. The subspace pooling is proved to be a special form of bilinear pooling [2]. The proof is referred to [4] for the detail.

**Accumulated stability [5]**.   The accumulated stability (AS) can be defined by $AS = \sum_{g \in G} \sum_{h \in G} |f(g) - f(h)|$ using the notation of our paper. The accumulated stability can be written as $(\sum_h |f(g) - f(h)|) \cdot \mathbf{1}$. It becomes a bilinear model when the output of the network $\alpha$ is $\sum_h |f(g) - f(h)| \in \mathbb{R}^{n_\alpha \times n_g}$ and the output of the network $\beta$ is all ones $\mathbf{1} \in \mathbb{R}^{n_g \times 1}$.

## 3   Architecture

We list the architectures of the models used in our experiments in Table 1, 2, 3, 4, 5, 6 and 7. In these tables, "Conv(output channels, kernel size, stride)" denotes a convolutional layer. "Linear(output channels)" denotes a fully connected layer. "AvgPool(kernel size,stride)" and "MaxPool(kernel size,stride)" denote a average pooling layer and a max pooling layer respectively. "Subspace-Pool(dim)" denotes a subspace pooling [4] which retains the first "dim" eigenvectors.

---

$^*$Corresponding authors: {xzhou,bao}@cad.zju.edu.cn. Project page: https://zju3dv.github.io/GIFT.

| VCNN | |
|---|---|
| **layer** | **operation** |
| conv0_sequential | Conv(32,5,1)-InstanceNorm-ReLU-AvgPool(2,2) |
| conv0_short | Conv(32,2,2)-InstanceNorm |
| conv0 = conv0_sequential + conv0_short | |
| conv1_sequential | Conv(32,5,1)-InstanceNorm-ReLU-AvgPool(2,2) |
| conv1_short | Conv(32,2,2)-InstanceNorm |
| conv1 = conv1_sequential + conv1_short | |
| conv2_sequential | Conv(64,5,1)-InstanceNorm-ReLU-AvgPool(2,2) |
| conv2_short | Conv(64,2,2)-InstanceNorm |
| conv2 = conv2_sequential + conv2_short | |
| conv3 | Conv(32,5,1)-InstanceNorm-L2Norm |

Table 1: Architecture of the Vanilla Convolutional Neural Network (VCNN).

| GFC | |
|---|---|
| **layer** | **operation** |
| Extractor | Conv(16,5,1)-InstanceNorm-ReLU- Conv(32,5,1)-InstanceNorm-ReLU-AvgPool(2,2) Conv(32,5,1)-InstanceNorm-ReLU- Conv(32,5,1)-InstanceNorm-L2Norm |
| Extractor($T_{g_i} \circ I$) | |
| fully connected | Linear(32*5*5, 512)-ReLU-Linear(512,128) |

Table 2: Architecture of Group Fully Connected Networks (GFC).

| GAS | |
|---|---|
| **layer** | **operation** |
| Extractor | Conv(16,5,1)-InstanceNorm-ReLU- Conv(32,5,1)-InstanceNorm-ReLU-AvgPool(2,2) Conv(32,5,1)-InstanceNorm-ReLU- Conv(32,5,1)-InstanceNorm-L2Norm |
| Extractor($T_g \circ I$) | |
| feature_network | Conv(64,1,1)-ReLU-Conv(128,1,1) |
| attention_network | Linear(800,512)-ReLU-Linear(512,25)-SoftMax |
| Sum(attention×features) | |

Table 3: Architecture of Group Attention Selection Networks (GAS).

| GIFT | |
|---|---|
| **layer** | **operation** |
| Extractor | Conv(16,5,1)-InstanceNorm-ReLU-<br>Conv(32,5,1)-InstanceNorm-ReLU-AvgPool(2,2)<br>Conv(32,5,1)-InstanceNorm-ReLU-<br>Conv(32,5,1)-InstanceNorm-L2Norm |
| Extractor($T_g \circ I$) | |
| group_conv1 | Conv(8,3,1) |
| group_conv2 | Conv(16,3,1) |
| BilinearPool(group_conv1,group_conv2) | |

Table 4: Architecture of the proposed method GIFT-1.

| Max Pooling | |
|---|---|
| **layer** | **operation** |
| Extractor | Conv(16,5,1)-InstanceNorm-ReLU-<br>Conv(32,5,1)-InstanceNorm-ReLU-AvgPool(2,2)<br>Conv(32,5,1)-InstanceNorm-ReLU-<br>Conv(32,5,1)-InstanceNorm-L2Norm |
| Extractor($T_g \circ I$) | |
| group_conv | Conv(128,3,1)-MaxPool(5,5) |

Table 5: Architecture of the model using max pooling.

| Average Pooling | |
|---|---|
| **layer** | **operation** |
| Extractor | Conv(16,5,1)-InstanceNorm-ReLU-<br>Conv(32,5,1)-InstanceNorm-ReLU-AvgPool(2,2)<br>Conv(32,5,1)-InstanceNorm-ReLU-<br>Conv(32,5,1)-InstanceNorm-L2Norm |
| Extractor($T_g \circ I$) | |
| group_conv | Conv(128,3,1)-AvgPool(5,5) |

Table 6: Architecture of the model using average pooling.

| Subspace Pooling | |
|---|---|
| **layer** | **operation** |
| Extractor | Conv(16,5,1)-InstanceNorm-ReLU-<br>Conv(32,5,1)-InstanceNorm-ReLU-AvgPool(2,2)<br>Conv(32,5,1)-InstanceNorm-ReLU-<br>Conv(32,5,1)-InstanceNorm-L2Norm |
| Extractor($T_g \circ I$) | |
| SubspacePool [4] | Conv(16,3,1)-SubspacePool(8) |

Table 7: Architecture of the model using subspace pooling