[Reviews · NeurIPS 2019]

Reviewer 1



The goal of this work is to compute descriptors for image keypoints that are invariant to a group of transforms -- yet retain distinctiveness. The authors attain this in a novel way, by treating image pixel descriptors as functions over the transform group. The effect is that when a transform is applied to an image, the function values at corresponding keypoints are simply permuted. The authors then apply group convolutions to the function values to aggregate local structure in these values over group neighborhoods in a way that retains equivariance, and finally they use two CNNs and group bi-linear blending to produce an invariant but discriminative descriptor that is sensitive to second-order statistics of the features. Overall this is a very nice generalization of the traditional CNN image pipeline, where equivariant convolutions over the translation group are then made invariant through max pooling. The paper does not describe at all (1) how edge effects are handled (e.g., padding rotated and down-scaled images), (2) how to guarantee that the group of transformations can be made compact so as to enforce the permutation property (otherwise function values can fall of the ends), and (3) limitations from the restriction that the group structure must itself be a regular grid, so that traditional convolutions can be applied (e.g., the authors consider only a discrete set of scales and rotations and their compositions). The discussion of prior work evaluation of the method, and ablation study, are good. This reviewer has read the authors' rebuttal. After the ensuing discussion, the final score assigned remains unchanged.

Reviewer 2



The submission introduces GIFT, a network that aims to learn image feature descriptors which are invariant to affine image transformations. Pros: + Extension of CNNs to aggregate descriptors computed from affine-transformed images. Cons: - Insufficient evaluation. - Lack of clarity. Details: While I think the extension of CNNs to learn descriptors from different transformed images is interesting and solid, I still have major concerns: Insufficient experiments. - While I value that the experiments use mainly planar scenes, I think the evaluations are missing experiments on more challenging datasets (e.g., a structure-from-motion dataset). At the end of the day, these descriptors mostly end up being used for camera relocalization in SLAM systems, 3D reconstruction, and image-based localization where scenes violate the assumption of smooth 3D surfaces in the scene. - The evaluations are missing experiments demonstrating robustness to systematic increases in rotation, scale, and view point. - I think experiments must show the std. deviation of the results shown in most Tables. This is because the results are aggregates from different images. This is important since not all images return the same number of keypoints; images with rich textures can return more keypoints than images with low-complex textures (e.g., walls). - Experiments are missing more details or lack clarity. In the experiments described in Section 4.3, what keypoint detector was used for the experiments? According to lines 276-277 the same keypoint detector was used. I assume these were Superpoint, DoG, and LF-Net shown in Table 4? If so, please make it explicit in the text. Lack of clarity. The submission is hard to read and clarity is key for reproducibility. In particular, Section 3 defines too many symbols and operations and barely relates them in Fig. 1. I think Fig. 1 should include some symbols and operations indicating at a high level the sequence of operations and the input and output they produce. By doing so, I think the submission can significantly improve clarity. Also, I think Fig. 2 is taking too much real state and informs too little. Lemma 1 is good enough to make the point that features from different warps are just "permuted" in the activation tensors. Another part that needs more clarity is in Eq. 2. What does it mean to construct matrix W defined on H? Does it mean there is a matrix W for every transformed h in H? Also, I assume W_i(h) is a column of the matrix W(h) and therefore b is a scalar. In sum, I think clarifying the dimensions of each element in Eq. 2 can improve clarity. Minor comments: - Use \max in Eq. 5. Post Rebuttal: The rebuttal addressed some of my concerns, but unfortunately triggered more questions than answers: 1. The rebuttal still included experiments mainly on planar scenes. These planar scenes will comply with the smooth assumption. Unfortunately, real scenes are *not* smooth. Real scenes have objects with complex geometry that makes the assumption of smoothness unrealistic. Most of the references that the rebuttal mentioned in the "Importance of affine-invariant descriptors." paragraph in the rebuttal use datasets with planar scenes. Adding results on scenes with more complex geometry (e.g., SfM datasets) would have added more interesting benefits and limitations of the approach. I would recommend checking out the following CVPR evaluation paper: Comparative Evaluation of Hand-Crafted and Learned Local Features, by J. Schonberger, et al.. 2. Why was SIFT not a baseline in the experiments measuring robustness to systematic increase in rotation and scale? 3. The experiments on SUN3D are not that challenging since the dataset mainly contains indoor scenes which clearly have too many dominant planes. As such, this dataset satisfies their assumption. But how about outdoor SfM datasets? 4. It is not clear how the experiments measure pose errors and how the experiments estimate the relative pose. I don't understand why the results only show rotation errors and not translation errors. 5. The rotation error from SIFT correspondences seems very large in my experience. Why is this? 6. The large number of inliers for GIFT+Dense may be an artifact of using a dense grid. If so, the comparison is not fair and we cannot draw a confident conclusion. 7. The experiments do not indicate how they define an inlier and what thresholds they used. Given these questions, my rating does not change.

Reviewer 3



Originality: 1. The paper leverages a group feature extraction process, where the output group features is proved to be equivariant to transformations. 2. The paper designs a novel group feature embedding module with two group CNNs and bilinear pooling. Quality: While I appreciate the authors' effort, one missing part is the motivation/rationale for using two group CNNs and bilinear pooling. A naive solution is to use one group CNN + fc layer to generate a descriptor. We could double the group CNN parameters to achieve a similar expressive power as the proposed group feature embedding module. Clarify: The paper is well-written. Significance: The paper shows improved performance on standard datasets. However, some important baseline comparison is missing. One baseline is to use one group CNN + fc layer, as described above. One is the Group Equivariant CNN, which is considered the most related work in the paper. Also, would a more advanced backbone feature network help, e.g., ResNet-101?

[Author Response · NeurIPS 2019]

We thank the reviewers for their constructive feedback. Our answers to all the questions are presented below.

## Reviewer 1

**Edge effects.** We use zero-padding when rotating an image. Downsampling doesn't need padding.

**Compact group.** Since the scale group is unbounded, in implementation we empirically select a reasonable range of scales and set feature values outside the range to zero. Thus, the permutation property doesn't rigorously hold near the boundary of the selected range. But empirical results show that this boundary effect will not obviously affect the final matching performance if the scale change is not too large.

**Limitations of regular grid.** We agree that regular grids and regular convolutions are only applicable to groups on which unit transformations can be defined. On groups without properly-defined unit transformations, we may resort to other techniques to compute group convolutions, e.g. G-FFT proposed in [9]. Provided well-defined group convolution, we can still exploit structures of group features to construct GIFTs.

## Reviewer 2

**Importance of affine-invariant descriptors.** If the observed object is smooth, the perspective transformation of a local region can be well approximated by an affine transformation [38]. Most of the existing works about local descriptors [3, 7, 11, 13, 21, 32, 35, 38, 40, 51, 56] focus on affine transformations or the subset of rotation and scaling. To the best of our knowledge, GIFT is the first CNN-based descriptor with provable robustness to rotation and scaling.

**Evaluation on datasets with systematic increase of scaling and rotation.** The following figure shows the performance of GIFT on the HPatches dataset with systematic increase of scaling and rotation. It shows that, with the increase of scale and rotation, the performance of GIFT degrades much more slowly than the performance of Superpoint.

|  | error(°) | inlier |
|---|---|---|
| SIFT | 35.88 | 104.97 |
| Superpoint | 20.83 | 60.48 |
| GIFT-SP | 17.02 | 65.66 |
| GIFT-Dense | **15.14** | **927.31** |

Table 1: Relative pose estimation.

|  | PCK | Std |
|---|---|---|
| GIFT | 67.15 | 14.59 |
| Superpoint | 53.66 | 21.88 |

Table 2: Standard deviation on View-HP.

**Experiments on more challenging datasets.** We have evaluated GIFT on a non-planar indoor dataset SUN3D using the PCK as the metric in Section 4.3. In order to further demonstrate the potential of GIFT for real computer vision tasks, here we provide additional results for relative pose estimation on the SUN3D dataset. The mean error of estimated poses and the average number of inlier correspondences are listed in Table 1. GIFT-SP uses SuperPoint as the detector while GIFT-Dense uses a dense grid as keypoints.

**Standard deviation.** The standard deviations on the View-HP dataset are given in Table 2. We will add the standard deviations for all other experiments in the revised manuscript.

**Clarity.** We thank the reviewer for the suggestions and will definitely improve the clarity in the revision.

(1) In Table 4 of Section 4.3, the first row lists names of the detectors. More explanations will be added in the text.

(2) We will add symbols to Figure 1 in the revised manuscript, as suggested by the reviewer.

(3) About Equation 2, $H$ contains 9 elements in the implementation. On every element $h \in H$, $W_i(h)$ is a vector with $n_{l-1}$ elements where $n_{l-1}$ is the number of input channels and $i$ is the index of output channels. The group convolution defined in Equation 2 was originally proposed in [8] as mentioned in Line 89-91. Due to the space limit, we only give a brief introduction and refer the readers to [8] for more details.

## Reviewer 3

**Motivation of using bilinear pooling.** As shown by Lemma 1 and Lemma 2, the transformation of an image results in a permutation of its group features. Thus, instead of using a permutation-sensitive FC layer, we adopt a permutation-invariant pooling operator to gain the invariance to transformations. We choose bilinear pooling rather than max or average pooling for two reasons. First, bilinear pooling collects expressive second-order statistics, retaining the distinctiveness of the resulting descriptors. Second, bilinear pooling makes GIFT a generalized model of former descriptors [21, 51, 56], as proved in the supplementary material. The ablation study in Table 2 shows that the bilinear pooling gives a better performance than max or average pooling.

**Comparison to the model with FC layers and Group Equivariant CNNs.** In Table 1 of the manuscript, we have compared GIFT-1 to GFC which uses FC layers instead of group convolutions. Here, we additionally provide the results of GFC with one more group convolution layer (GFC+GC) in Table 3. The original Group Equivariant CNN is not designed for this task and is not directly comparable. Instead, we have implemented a baseline model similar to Group Equivariant CNNs, which is described in Line 252-255, and the results are reported in Table 2 of the manuscript (max pooling).

|  | ER-HP | ES-HP |
|---|---|---|
| GIFT-1 | 39.68 | 21.74 |
| GFC+GC | 30.25 | 17.23 |

Table 3: PCK on ER- and ES-HP.

**Advanced backbones.** Advanced backbones with more training data may help. But our objective is to improve invariance with properly designed geometric components which can be integrated in any CNN backbones .

[Meta-Review · NeurIPS 2019]

This paper generated some disagreement among reviewers. The main strength is the novel technical contribution which can inspire others along similar directions. The main weakness is the lack of experimental evaluations that were suggested by the reviewers (additional datasets, baselines, and ablations). We hope that these can be included for the final revision.